# Changing Landscape of Systemic Therapy in Biliary Tract Cancer

**DOI:** 10.3390/cancers14092137

**Published:** 2022-04-25

**Authors:** Edward Woods, Dat Le, Bharath Kumar Jakka, Ashish Manne

**Affiliations:** 1Department of Internal Medicine, The Ohio State University College of Medicine, Columbus, OH 432120, USA; edward.woods@osumc.edu; 2Department of Pharmacy, The Arthur G. James Cancer Hospital and Richard J. Solove Institute at The Ohio State University, 460 W 10th Ave, Columbus, OH 43210, USA; dat.lle@osumc.edu; 3Department of Internal Medicine, Baptist Medical Center South, Montgomery, AL 36116, USA; bharath_jakka@teamhealth.com; 4Department of Internal Medicine, Division of Medical Oncology at the Arthur G. James Cancer Hospital and Richard J. Solove Research Institute, The Ohio State University Comprehensive Cancer Center, Columbus, OH 43210, USA

**Keywords:** cholangiocarcinoma, gall bladder cancer, FGFR2, pemigatinib, infrigatinib, HER2, durvalumab, gemcitabine, NTRK, IDH

## Abstract

**Simple Summary:**

Cancers from the bile ducts and gall bladder are lethal. Cure by surgery is not possible in most of these tumors as they are often identified in later stages. Unlike other cancers, they have few chemotherapy treatment options. Multiple clinical trials in the past decade failed and could not replace the combination of two drugs, gemcitabine and cisplatin, as the preferred drug in new cases. Patients who fail to respond to this combination do not have reliable treatment options. The success of therapy directed at a specific genetic change (mutation) and activating the patient’s immune system (alone or in combination with chemotherapy) are encouraging. If we follow the current trials, the focus is on these newer treatments, with there being a high chance they may replace traditional chemotherapy in the future.

**Abstract:**

Biliary tract cancers (BTC) are often diagnosed at advanced stages and have a grave outcome due to limited systemic options. Gemcitabine and cisplatin combination (GC) has been the first-line standard for more than a decade. Second-line chemotherapy (CT) options are limited. Targeted therapy or TT (fibroblast growth factor 2 inhibitors or FGFR2, isocitrate dehydrogenase 1 or IDH-1, and neurotrophic tyrosine receptor kinase or NTRK gene fusions inhibitors) have had reasonable success, but <5% of total BTC patients are eligible for them. The use of immune checkpoint inhibitors (ICI) such as pembrolizumab is restricted to microsatellite instability high (MSI-H) patients in the first line. The success of the TOPAZ-1 trial (GC plus durvalumab) is promising, with numerous trials underway that might soon bring targeted therapy (pemigatinib and infrigatinib) and ICI combinations (with CT or TT in microsatellite stable cancers) in the first line. Newer targets and newer agents for established targets are being investigated, and this may change the BTC management landscape in the coming years from traditional CT to individualized therapy (TT) or ICI-centered combinations. The latter group may occupy major space in BTC management due to the paucity of targetable mutations and a greater toxicity profile.

## 1. Introduction

Biliary tract cancers (BTC) comprise a group of malignancies originating in the epithelium of the biliary tract [1]. These include cholangiocarcinoma (CCA) and gallbladder carcinoma (GBC). Intrahepatic cholangiocarcinoma or iCCA refers to tumors proximal to the second-order ducts, while extrahepatic cholangiocarcinoma or eCCA refers to tumors arising more distally (perihilar CCA, between second-order ducts and cystic duct and distal CCA, distal to cystic duct) [2]. Perihilar CCA represents 50% of the total CCAs, with distal lesions comprising 40% and the final 10% being intrahepatic [3]. BTCs are relatively rare in developed countries, comprising approximately 3% of gastrointestinal malignancies with an incidence of 0.35 to 2 in 100,000 [4]. In developing countries such as China and Thailand, the incidence can be as high as 14–80 in 100,000. GBCs are less common, with an incidence of 1 in 100,000 in the USA but increasing as high as 27 in 100,000 in Chile [5,6]. Risk factors for CCAs include primary sclerosing cholangitis, choledochal cysts, cholelithiasis, hepatolithiasis, chronic liver disease, genetic conditions such as Lynch syndrome, BRCA mutations, cystic fibrosis, biliary papillomatosis, and liver fluke infection in endemic regions [7,8]. Risk factors for GBC include cholelithiasis, chronic infection with pathogens such as salmonella and Helicobacter pylori, obesity, and anatomical changes in the biliary tree [9]. The continued rise of CCAs, specifically iCCA, in the past four decades globally is concerning [10,11,12]. Its association with metabolic and infectious risk factors might be the primary reason for this dangerous trend.

A lack of robust screening measures, late diagnosis (unresectable to metastatic), challenging histology at presentations combined with limited systemic options, the high recurrence rate after surgery, and unreliable biomarkers to monitor the treatment response contribute to poor outcomes in BTCs [13]. Surgical management is curative in early-stage BTC, but it is feasible in only a small fraction of cases (≈30%) [14,15]. Therefore, the majority of the patients must be treated with systemic therapy and palliative intent. Even with resection, 3-year recurrence rates can be as high as 80% [16]. Liver transplant is approved for certain unresectable hilar or perihilar eCCA (≤3 cm, absent nodal and intra or extrahepatic metastatic disease and no biopsy) only [17].

This literature review discusses systemic therapy’s current chemotherapy-centric landscape with a limited role of targeted therapy and immune checkpoint inhibitors (ICI). The status of newer targets and newer agents for established targets, ICI-based combinations on the horizon, and their impact on shaping the future of BTC management is also examined. The emphasis of this paper will be on palliative therapy. The adjuvant therapy (AT) and neoadjuvant therapy (NAT) options will be briefly discussed.

## 2. Chemotherapy in Biliary Tract Cancers

### 2.1. Chemotherapy in the First Line

Over 70% BTCs present in advanced stages or aBTC (unresectable or metastatic) and are only eligible to receive palliative therapy. The combination of gemcitabine (Gem) and cisplatin (Cis), or GC, is the current approved first-line therapy [18]. There were no positive first-line trials for over a decade. The standard approach to BTCs is illustrated in Figure 1.

In ABC-01, a phase II randomized trial, GC combination was compared to Gem alone in treatment-naïve aBTC patients [19]. The tumor response rates (28% vs. 23%), time to progression (8 months vs. 4 months), and 6-month progression-free survival or PFS rate (57% vs. 46%) were higher in the combination group. GC approval in the first line was based on the ABC-02 trial, a phase III randomized control trial in which GC was compared to Gem alone. The median overall survival or OS (11.7 months vs. 8.1 months; hazard ratio or HR = 0.64; *p* < 0.001) and the median PFS (8 months vs. 5 months; HR = 0.63; *p* < 0.001) was higher in the GC group. The tumor control (complete response (CR) or partial response (PR) or stable disease (SD)) was also higher in the GC group (81% vs. 72%; *p* = 0.04). The tolerance profile was comparable between both groups, except for neutropenia (higher with GC). More details are discussed in Appendix A.

The combination of oxaliplatin, irinotecan, and infusional fluorouracil (mFOLFIRINOX) was inferior to GC in the first-line setting, as evidenced by the PRODIGE 38 AMEBICA trial [20]. In this randomized phase II/III trial, the 6-month PFS rate (44.6% in mFOLFIRINOX vs. 47.3% in GC), PFS (6.2 m vs. 7.4 m), and OS (11.7 m vs. 13.8 m) were superior in the GC group. A partially activated monophosphorylated Gem compound, NUC-1031, that can overcome the resistance developed against Gem, was tested in the first line for aBTC [21]. This compound does not need a nucleoside transporter to enter the cell, has enzyme-mediated activation, and resists degradation by cytidine deaminase [22]. Although early trials with NUC-1031 plus Cis had a greater objective response rate or ORR over GC (44% vs. 26%), the phase III trial was discontinued as the interim analysis showed that it would be unlikely to meet its primary end-point of 2.2 months superiority in OS compared to GC [21]. In the BREGO trial, Regorafenib (Reg) and GEMOX (gemcitabine and oxaliplatin combination) were compared to GEMOX alone in aBTC [23]. The overall results were unsatisfactory (the Reg-GEMOX group was not superior to the GEMOX-only group for PFS or OS). Subgroup analysis showed a higher disease control rate (or DCR), PFS, and OS in patients who continued Reg beyond four cycles.

The addition of nab-paclitaxel (NP) to GC (GC/NP) in the first line had encouraging results in a single-arm phase II trial [24]. The hematological toxicity was very high in the first 32 (of 60) patients enrolled in the trial who received Gem (1000 mg/m^2^), Cis (25 mg/m^2^), and NP (125 mg/m^2^) on days 1 and 8 of 21-day cycles. The doses of Gem and NP were dropped to 800 and 100 mg/m^2^, respectively, for the next 28 patients. The median PFS was 11.8 months and the median OS was 19.2 months. DCR (PR plus SD) was superior in the high-dose group (90% vs. 78% in reduced dose). Comparing GC and GC/NP is not ideal (no head–head trials), but GC/NP seems to have a better OS and PFS, and worse neutropenia and anemia, based on observations from the respective published trial data (please refer to Appendix A for more details) [18,24].

In a Korean retrospective review from four medical centers, the safety and efficacy of GC/NP in treating aBTC was reported last year [25]. The authors looked at the outcomes (ORR, DCR, PFS, and OS) in two groups of patients based on when they received GC/NP: a) in the first line; b) NP was added to GC before or after disease progression (PD). The former group’s ORR (48% vs. 31%) and DCR (90% vs. 75%) were superior. The ORR (40% vs. 16%) and DCR (86% vs. 60%) were greater when NP was added before PD in the latter group. The safety profile was acceptable in these patients and, as expected, Grade 3/4 events were lower in patients who received a reduced dose of GC/NP. A phase III randomized trial (SWOG1815, NCT03768414) is underway to examine the benefit of adding NP to GC in aBTC (GC/NP vs. GC). GC plus S-1 (an oral fluoropyrimidine derivative) combination has a survival benefit over GC in treating aBTCs [26]. The preliminary data of KHBO1401-MITSUBA, a phase III randomized trial, showed improved OS (13.5 months vs. 12.6 months), PFS (7.4 months vs. 5.5 months), and response rates (41% vs. 15%) in the triplet group compared to the GC group.

In the TOPAZ-1 trial, phase III randomized, double-blind, placebo-controlled GC plus durvalumab (ICI) or GC-D was compared to GC plus a placebo [27]. Patients received GC-D for eight cycles (days 1 and 8, Q3W) followed by durvalumab only or placebo Q4W. The mOS 12.8 months vs. 11.5 months (hazard ratio [HR], 0.80; 95% confidence interval [CI], 0.66–0.97; *p* = 0.021), mPFS 7.2 months vs. 5.7 months (HR, 0.75; 95% CI, 0.64–0.89; *p* = 0.001), and ORR (26.7% vs. 18.7%) was superior in GC-D compared to the GC group. G3/4 AEs were similar in both groups. While the results of the GC-D combination are promising, we need to wait for the full study data to make reliable conclusions. The results of other clinical trials are discussed in Table 1. Although it is not ideal to compare the results from the ABC-02, TOPAZ-1, and GC/NP trials, we attempted to compare the survival data and toxicity profile (of few AEs) in Appendix A.

### 2.2. Chemotherapy in the Second Line

In aBTC (and ampullary cancers), patients who progressed on GC with a preserved performance status (Eastern Cooperative Oncology Group or ECOG scale of 0–1), FOLFOX had a small OS benefit (6.2 months vs. 5.3 months; adjusted hazard ratio = 0.69 [95% CI 0.50–0.97]; *p* = 0.031) compared to supportive care [28]. The survival rate was higher in the FOLFOX group at 6 months (51% vs. 36%) and 1 year (26% vs. 11%). Subgroup analysis in this trial produced some interesting results. The OS (not PFS) was superior with FOLFOX among the platinum-sensitive (PD after 90 days of completion of first-line chemotherapy) and platinum-resistant/refractory (PD on the first line or in less than 90 days after completion of first-line chemotherapy). Expectedly, high-grade AE were more prevalent in the FOLFOX group (69% vs. 52%). A retrospective study in Italy examined the differences in outcomes after second-line chemotherapy (post-GC) between elderly (≥70 years) and younger (<70 years) patients. There were no significant differences in the outcomes (OS or PFS) between the two groups. The most-used second-line agents in the elderly population were Gem alone or capecitabine alone or a combination of both. Treatment-related toxicity was very high in the elderly population compared to the younger group (48.5% vs. 8.2%; OR 6.31; *p* < 0.001) [29].

A combination of nanoliposomal irinotecan (Nan-Iri) and 5FU was compared to 5FU alone in the NIFTY trial [30]. It was a multicenter, open-label, randomized, phase IIb trial in which patients progressed on GC. The combination group had a superior PFS (7.1 m vs. 1.4 m; HR = 0.56; 95% CI 0.39–0.81; *p* = 0.0019) and ORR (19.3% vs. 2.1%) compared to the 5FU group. G3-4 neutropenia (24% vs. 1%) and serious adverse events (42% vs. 24%) occurred more in the combination group than the 5FU-only group. It was concluded that Nan-Iri plus 5-FU could be considered for second-line treatment in patients with BTC who formerly progressed on GC, especially in patients who cannot tolerate platinum agents. On the other hand, mFOLFIRINOX had reasonable efficacy and safety for patients who progressed on GC (≥3 cycles) and is an option for patients with no targetable mutations [31].

## 3. Targeted Therapy in Biliary Tract Cancers

Second-line options in patients who progressed on GC are limited. In the subset of patients with targetable mutations, fibroblast growth factor 2 (FGFR2) inhibitors such as those with pemigatinib and infrigatinib [32], neurotrophic tyrosine receptor kinase (NTRK) gene fusions such as larotrectinib and entrectinib [33,34], and isocitrate dehydrogenase 1 (IDH-1) with ivosidenib [35], are suitable agents which are preferred over chemotherapy in the second line (preferably after GC). Individual targeted therapy options will be discussed in the following text. The reported results of trials and ongoing trials with targeted therapy are summarized in Table 1 and Table 2.

**Table 1 cancers-14-02137-t001:** Results of recent trials in biliary tract cancer.

Line	Phase (N)	Clinical Trial Identifier	Treated Cancer Group	Experimental Arm	Target of the Drug (If Applicable)	Comparative Arm	Primary Outcome Studied in the Trial	Top 3 Treatment-Related Adverse Events	Notes
First line	III	NCT03875235 [27]	BTC	Durvalumab (D) + GC	PD-1	GC + placebo (Pbo)	OS—12.8 m vs. 11.5 m (D vs. Pbo, HR = 0.80; 95% CI, 0.66–0.97; *p* = 0.021)	Anemia Low neutrophil count Low platelet count	PFS-7.2 m vs. 5.7 m (D vs. Pbo, HR, 0.75; 95% CI, 0.64–0.89; *p* = 0.001); ORR—26.7% vs. 18.7% (D vs. Pbo); Grade 3/4—62.7% vs. 64.9% (D vs. Pbo)
	II	NCT03796429 [36]	BTC	Toripalimab + GC	PD-1	Single arm	PFS—6.7 m OS—NR	Leukopenia Anemia Rash	ORR—21 DCR—85% G3/4, non-hematological in 20% and hematological—69%
	II	NCT03951597 [37]	iCCA	Toripalimab + lenvatinib + GemOx +	PD-1 + TKI	Single arm	ORR—80% (1CR and three patients obtained enough control to allow for resection)	Jaundice Rash Proteinuria	DCR—93.3%, PFS—10 m OS—NR DOR—9.8 m
	II	NCT04361331 [38]	iCCA	Lenvatinib + GemOx	TKI	Single arm	ORR—30% 1/30 was down staged to have resection	Fatigue Jaundice Vomiting	PFS and OS—NR DCRc—87% No G5, ≥G3 in 40%
	Ib II	NCT02992340	BTC	Varlitinib + GC	Pan-HER 2	Single arm	DLT—1/11 (200 mg); 1/12 (300 mg)	blood and lymphatic system disorders	PR = 8/23; SD = 12/23 ORR—35%, DCR—87%, DoR—4 m, PFS—6.8 m
	Ib II	NCT02128282 [39]	CCA	Silmitasertib (CX-4945) + GC	Casein kinase 2 (CK2)	Single arm	PFS 11 m	Diarrhea Neutropenia Nausea	Compared to GC—Better PFS Lesser neutropenia
	I	NCT02375880 [40]	BTC	DKN-01 + GC	Dickkopf-1 (DKK1)	Single arm	Safety—no DLT	Neutropenia Thrombocytopenia Leukopenia	ORR—21.3% PFS—8.7 m
Subsequent lines	III	NCT02989857 (ClarIDHy) [41]	CCA	Ivosidenib (IVO)	IDH-1	IVO alone vs. placebo	PFS—2.7 m vs. 1.4 m (HR = 0.37; 95% CI 0.25–0.54; *p* < 0.0001).	Ascites Fatigue Anemia	OS in updated analysis 10.3 m IVO vs. 7.5 m (HR = 0.79; 95% CI 0.56–1.12; *p* = 0.093)
	II	NCT02966821 [42]	BTC	Surufatinib	VEGF	Single arm	PFS rate at 16 wks—46.33% (95%, 24.38–65.73)	Elevated bilirubin Hypertension Proteinuria	PFS—3.7 m OS—6.9 m
	II	ChiCTR1900022003 [43].	BTC	Anlotinib + sintlimab	TKI + PD-1	Single arm	OS—NR	Hypertension ** Diarrhea Hypothyroidism	PFS—6.5 m ORR—40% DCR—87%
	II	NCT02052778 [44].	iCCA #	Futibatinib	FGFR2	Single arm	ORR 37%	Hyperphosphatemia Diarrhea * Dry mouth *	DoR—8.3 m and DCR = 82%
	II	NCT03230318 [45]	iCCA	Derazantinib	FGFR2—mutations and amplifications	Single arm	3-month PFS rate—76%	Not specified	DCR = 80% PFS = 7.3 m 6-month PFS rate = 50%
	II	NCT03797326 [46]	BTC #	Pembrolizumab + lenvatinib	PD-1 + TKI	Single arm	ORR—10% Safety—TRAE in 97% (>G354%)	Hypertension Dysphonia Diarrhea	DCR—68% PFS—6.1 m OS—8.6 m
	II	NCT02265341 [47]	BTC	Ponatinib	FGFR2	Single arm	ORR—9%	Lymphopenia, Rash Fatigue (50%)	CR = 0, PR—8%, SD = 36%. PFS—2.4 m and OS—15.7 m
	II	NCT03834220 [48]	CCA among Solid tumors	Debio 1347	FGFR Fusion	Single arm	ORR—2/5 (40%) of CCA	Fatigue Hyperphosphatemia Anemia	DoR and PFS were 16.1 weeks and 18.3 weeks (in all patients), respectively.
	II	NCT01953926 [49]	BTC + AC #	Neratinib	HER2 or EGFR Exon 18	Single arm	ORR—12%	Diarrhea * Vomiting *	PSS—2.8 m OS—5.4 m
	I/ II	NCT01752920 [50]	iCCA	Derazantinib	FGFR2—fusions	Single arm	Safety—all-grade TRAE in 93%	Fatigue Eye-toxicity Hyperphospatemia	≥3 Grade TRAE in 28% ORR—27% DCR—83%
	I	NCT02699515 [51]	BTC #	Bintrafusp alfa,	TGF-β and PD-L1	Single arm	Safety—emergent and all adverse events	Rash Fever Increased lipase	63% had TRAE 37% ≥ G3
	I	NCT02892123 [52]	BTC #	ZW25 (Zanidatamab)	bispecific HER2	Single arm	Safety/tolerability—only G1–G2 reported in 70%	Fatigue ** Diarrhea Infusion reaction	ORR—47 DCR—65% DoR—6.6 m
	Ib	NCT03996408 [53]	BTC	Anlotinib TQB2450	TKI + PDL1	Single arm	DLT/ MTD in first 3 weeks (one cycle)—none RP2D—25 mg ORR—42%	* Hypertension Leukopenia Increased total bilirubin Neutropenia	PFS—240 days DCR—75%

# Part of a basket trial but these results are from the BTC cohort; * All grade AE, ** G1-G2 AE; BTC—biliary tract cancers include gall bladder cancers and CCA; iCCA—intrahepatic cholangiocarcinoma; eCCA—extra-hepatic cholangiocarcinoma; CCA—cholangiocarcinoma includes iCCA and eCCA; AC—ampullary cancer; GC—gemcitabine/cisplatin; Gem/Ox—gemcitabine/oxaliplatin; OS—median overall survival; PFS—median progression free survival; m—months; wks—weeks; HR—hazard ratio; CI—confidence interval; TRAEs—treatment-related adverse events; NR—not reached; DCR—disease control rate; ORR—objective response rate; CR—complete response; PR—partial response; DOR—duration of response; IDH—isocitrate dehydrogenase-1; VEGF—vascular endothelial growth factor; FGFR2—fibroblast growth factor 2; HER2—human epidermal growth factor receptor 2 inhibitors; EGFR—epidermal growth factor receptor; mab—monoclonal antibody; TGF—transforming growth factor; PD-1—programmed cell death protein 1; PDL1—programmed cell death ligand protein; TKI—tyrosine kinase inhibitor; DLT—dose limiting toxicity; MTD—maximum tolerated dose; R2PD—recommended phase II dose.

### 3.1. Fibroblastic Growth Factors Receptor Inhibitors (FGFRis)

Fibroblast growth factors (FGF) are protein ligands that play a vital role in regulating cell proliferation, differentiation, migration, and tissue repair/angiogenesis [54]. FGFRs are transmembrane proteins with three extracellular domains (D1–D3), a transmembrane domain, and an intracellular tyrosine kinase domain 13 [55]. There are 18 types of FGF (FGF1–10 and 16–23) that can bind to a family of 4 FGFRs (FGFR1–4) [56]. The effects of ligand (FGF) binding on FGFR can be simplified as follows (in order): dimerization of FGFR, transphosphorylation of TK domains, attachment of which adaptor proteins at the phosphorylated site (docking site), phosphorylation of adaptor proteins, activation of a cascade of downstream signaling pathways, Ras-Raf-MAPK, PI3K-AKT, Stat, and PLCγ, and gene transcription [57]. Any alterations in the FGFRs, such as amplification, mutation, and fusion/rearrangement, can activate the above-mentioned pathway constitutively, promoting uncontrolled cell growth, migration, and survival, ultimately leading to malignant transformation [58].

The prevalence of FGFR alterations among solid tumors (tissues) is approximately 7% and, when detected, are common in the lung, colon, breast, endometrial adenocarcinoma, and glioblastoma multiforme [59,60]. The majority were in FGFR1 (49%), followed by FGFR3 (26%), FGFR2 (19%), and FGFR4 (7%) [60]. Approximately 5% of tissues had >1 FGFR alterations. When classified by the kind of alterations, two-thirds had amplifications, a quarter had mutations, and only 8% had fusions. The frequency of FGFR fusions is greater in CCA, specifically in iCCA (14%), compared to other solid tumors (colorectal, hepatocellular and gastric) [61].

Among the FGFR alterations, FGFR2 fusions/rearrangements have a favorable prognostic impact (even with chemotherapy) in BTCs and are more sensitive to FGFR inhibitors (FGFRi), as reported in retrospective studies [58,62,63]. Currently, the indication for using pemigatinib and infrigatinib (FGFRis) is BTC patients with FGFR2 fusion/rearrangement who progressed on chemotherapy (GC) [64,65]. FLIGHT-202 is a single-arm phase II trial where CCA patients with FGFR2 rearrangement or fusions were treated with pemigatinib in the second line (N = 107) [64]. Although there were small cohorts of patients with other (N = 20) or no (N = 18) FGF/FGFR alterations, the primary objective was to study the ORR in patients with fusions/rearrangements. With an ORR of 36% (CR in 3%, PR in 33%, SD in 47%), DCR of 82%, a 1-year PFS rate, and an OS rate of 29% and 68%, pemigatinib earned approval as an ideal second-line agent for patients with FGFR fusions/rearrangements. Alternatively, pemigatinib did not show any efficacy benefit in the other two cohorts (other or no FGF/FGFR alterations). Infrigatinib, a selective, ATP-competitive FGFRi, was also studied in a similar population [65]. The ORR was 25% (CR in 1%, PR in 22%, SD in 61%), DCR was 84%, and the median DOR was 5 months. The median PFS and OS were 7.3 months and 12.2 months, respectively. Both drugs could cause severe hyperphosphatemia requiring aggressive management. Fatigue, stomatitis, hyponatremia, palmar-plantar erythrodysesthesia syndrome, and alopecia are some of the other important AEs with these drugs.

Other FGFRis with early success include derazantinib (ARQ 087), futibatinib, and Debio1347 [50,66,67]. The interim results of BTCs with FGFR2 mutations or amplifications (not fusions/rearrangements) treated with dezaratinib in FIDES-01 trial were reported recently [45]. Of 28 patients enrolled in this trial, 78% had missense point mutations and the remainder were other short variants and amplifications. The DCR was 74% (PR in 8.7% and SD in 65%), with the PFS rate at 3 months and 6 months being 76% and 50%, respectively. The response was seen across all types of alterations. Erdafitinib, another FGFRi, showed durable responses (ORR of 41%, median DOR of 7.3 months) and an acceptable safety profile in CCA with FGFR fusions/rearrangements and mutations in the second line [68].

### 3.2. Isocitrate Dehydrogenase Inhibitors

Isocitrate dehydrogenase (IDH) is a key enzyme in the TCA (tricarboxylic acid) cycle and helps in converting isocitrate to α-ketoglutarate [69]. A mutant IDH-1 produces an abnormal enzyme that further converts isocitrate to α-ketoglutarate to a metabolite with malignant potential, 2-hydroxyglutarate (2-HG) [70]. The prevalence of IDH-mutations (IDH-1/IDH-2) is <5% among BTCs, while IDH-mutant tumors typically have lower tumor mutation burden (TMB) and rarely have microsatellite instability or PDL-1 positivity compared to IDH-wild type tumors [71,72,73].

Ivosidenib (IVO) is one of the first IDH-1 inhibitors that showed benefit in treating CCA [35]. The first phase III results (ClarIDHy) published last year showed a statistically significant improvement in the PFS (2.7 months in IVO vs. 1.4 months in placebo; HR = 0.37; *p* < 0.0001) in refractory CCAs treated with IVO (compared to placebo). This trial allowed crossover from placebo to IVO group after progression. Only 30% (vs. 22% in placebo) had drug-related serious AEs. An updated analysis of this trial reported higher OS in IVO group (10.3 months vs. 7.5 months; HR = 0.79; *p* = 0.09), but when adjusted to crossover (derived using rank-preserving structural failure time), there was a 2-month survival advantage in the IVO group (7.5 m in IVO vs. 5.1 in placebo; *p* = 0.0001) [41]. LY3410738 is a mutated-IDH1 inhibitor that is different from commonly used IDH-1 inhibitors such as ivosidinib in that it binds covalently to the mutant enzyme and at a different site, thereby reducing the risk of secondary mutations [74]. It is being studied in a phase I basket trial in the second line [75]. There is also a group for CCA where it is combined with GC.

### 3.3. Neurotrophic Tyrosine Receptor Kinase Fusion Inhibitors

Fusions in neurotrophic tyrosine receptor kinase (NTRK) genes that encode tropomyosin receptor kinases (TRK) promote carcinogenesis and were identified as driver mutations in many cancers, including BTCs [76]. The prevalence of NTRK fusions among BTCs is very low (0.75%) [77]. Successful basket trials gave two NTRK inhibitors, larotrectinib (Lt) and entrectinib (Et), that have very high response rates and are well-tolerated [33,34]. It should be noted that these basket trials had very limited CCAs in their study population (2/55 for Lt and 1/54 for Et).

### 3.4. Vascular Endothelial Growth Factor Inhibitors

The overexpression of vascular endothelial growth factor (VEGF) and vascular endothelial growth factor receptors (VEGFR) is common among BTCs (54% in CCA) and contributes to their poor outcomes [78,79]. In a phase II study (NCT02520141) published recently, the benefit of ramucirumab, a fully human, IgG1 monoclonal antibody direct inhibitor of VEGFR-2, was studied in treatment-refractory BTCs [80]. The response achieved was in line with other agents used in the similar population (PR in 2%, SD in 43%, mPFS of 3.2 months, and OS of 9.5 months). Suraftinib, a small-molecule inhibitor of VEGFR 1-3, was studied in the second-line (phase II) for BTC [42]. Patients received 300 mg daily in 28-day cycles. The primary end-point was a 16-week PFS rate that was 46.33% (95%, 24.38–65.73). The mPFS (3.7 months) and mOS (6.9 months) were reasonable. Interestingly, it was more effective in patients with disease in the liver and lower baseline CA 19-9 (≤1000 IU/mL). As expected, hypertension and proteinuria were frequent in the study population. A Chinese trial looked at the combination of a camrelizumab, ICI (anti-PD-1), and apatinib, a VEGFR2 inhibitor in the neoadjuvant setting for aBTC [81]. Efficacy, safety, and the exploration of biomarkers were the study’s aims. Patients received 200 mg carmelizumab/2 weeks with VEGFR2 inhibitor 250 mg/day for two cycles. Among 17 subjects (13 GBC and 4 CCA), the ORR was 71%, including 2 patients with CR. In a prospective trial with 22 patients with aBTCs, this combination had a DCR of 71% (PR—19% + 50% SD). G3/G4 TRAEs were reported in 64% of patients [82].

Anlotinib (AL3818) is a novel oral receptor tyrosine-kinase inhibitor (TKI) that works on VEGFR-2 and -3, FGFR1-4, platelet-derived growth factors (PDGFR)-α and -β, c-Kit and Ret, and inhibits tumor growth and angiogenesis [83]. It has shown some promise in lung cancer (NSCLC/SCLC), RCC, esophageal, and other solid tumors [84,85,86,87]. It was studied in combination with TQB2450, a PD-L1 inhibiting ICI in an open-label phase Ib trial in refractory aBTC [53]. Anolitinib at 12 mg dose with ICI was considered a safe dose. HTN, elevated bilirubin, and leukopenia/neutropenia were the top AEs. In evaluable patients, the ORR was 42% with DCR of 75%. In another phase II study, it was used in the second line along with another novel ICI, sintlimab (PD-1 inhibitor at 200 mg/Q3 weeks). When reported, primary end-point OS was not reached after a median follow-up of approximately 9 m. As was the case with other trials, HTN was seen in most patients (70%). The ORR (40%) was similar to other trials, but DCR was slightly higher (87%) [43].

### 3.5. Human Epidermal Growth Factor Receptor 2 Inhibitors

HER2 (human epidermal growth factor receptor 2) belongs to a family of four epidermal growth factor receptors (HER1–4) and has a proven role in malignancy when overexpressed or amplified [88]. When activated by dimerization, HER2 triggers phosphorylation of certain tyrosine kinases that promote cell growth/proliferation and malignant transformation through a series of downstream signaling pathways [88]. HER2 overexpression/amplification is seen in 3–20% of BTCs and is less common in iCCA (compared to eCCA and GBC) [89,90]. HER2 inhibitors are commonly used to treat HER2-positive breast, esophageal, and gastric cancers [91,92]. They are now being studied to treat aBTCs.

Trastuzumab, a humanized monoclonal antibody (hmab) directed against the extracellular domain (IV) of HER2, suppresses signaling pathways and degrades HER2 [93]. Pertuzumab, another hmab, alternatively prevents the dimerization of HER2 receptor by attaching to another extracellular domain (II) and inhibiting its activation [94]. Zanidatamab (ZW25) is a bi-specific antibody that binds to both domains II and IV [95]. Neratinib is an irreversible TKI that binds to the intracellular TK-domain and inhibits signaling pathways [96].

The results of the BTC expansion cohort of the phase I trial of zanidatamab (ZW25) were reported [52]. A total of 20 refractory BTC (including 5 patients who received prior-trastuzumab) were given 20 mg/Q2 weeks. No serious AEs were reported, while 70% had G1/2 AE. A BTC cohort that had 25 patients (including AC) reported for a SUMMIT trial, where refractory solid tumors were treated with neratinib alone (24 mg/day). [49]. The ORR (primary end-point) was appreciated in just 12%. Despite receiving loperamide prophylactically, 56% had diarrhea, including G3 in 24%, but the drug was not discontinued in any patient with diarrhea. GC with varlitinib (reversible pan-HER inhibitor) was well tolerated in treatment-naïve Asian patients [97]. A phase Ib trial determined the maximum tolerated dose of varlitinib and safety of the combination. Dose-limiting toxicity (DLT) was reported in 2/11 and 1/12 patients in 200 mg and 300 mg cohorts, respectively. The 300 mg cohort had a higher rate of ≥grade 3 AEs (67% vs. 36%). The ORR was 35% (PR in 35% and SD in 35%) during the study period.

Ongoing phase II trials with HER2 inhibitors in refractory BTC are: (a) Trastuzumab plus mFOLFOX in aBTC including AC; (b) Trastuzumab plus pertuzumab (part of a basket trial including My Pathway trial); (c) Zanidatamab monotherapy in aBTC; (d) Trastuzumab deruxtecan in a BTCs (DESTINY-PanTumor02, NCT04482309); (e) Trastuzumab emtansine (NCT02999672) [98,99,100,101].

### 3.6. Other Targeted Therapy Options

The ROAR and NCI-Match trials showed the ORR ranging from 20–38% with dabrafenib plus trametinib combination in BTCs with BRAFV600E mutation and may be an option if they progress on first-line GC [102,103]. Other BRAF inhibitor combinations under investigation are dabrafenib plus JSI-1187 (ERK-inhibitor) in NCT03272464 and ABM-1310 (novel BRAF-inhibitor) plus cobimetinib (MEK-inhibitor) in NCT04190628 (discussed in Table 2). The effect of encorafenib plus bimimetinib (MEK-inhibitor) on non-V600E BRAF mutations is being examined in a phase II trial (NCT03839342). GC in combination with selumetinib (MEK inhibitor) had acceptable toxicity (1/12 had DLT of chest pain) in an ABC-04, phase I trial [104]. Three patients had PR and 5 had SD. A phase II clinical trial comparing this combination (GC plus selumetinib) to GC in aBTC (NCT02151084) is now underway.

Protein kinase CK2 is a phosphorylating enzyme that is essentially active in the normal eukaryotic cell that helps in cell differentiation and immune regulation [105,106]. It has a role in benign diseases such as diabetes, neurological disorders such as Parkinson’s disease and Huntington’s disease, intestinal inflammation, and some autoimmune diseases [105,107,108]. Its role in malignant transformation, distant metastasis, and drug resistance is well established [109]. Silmitasertib (CX-4945) is a selective inhibitor of CK2 with antiproliferative/antiangiogenic capability that showed good efficacy in preclinical studies [110]. In a multicenter, open-label, phase Ib/II study, it was administered along with GC in unresectable CCA (1000 mg bid, 10 days of 21 days GC cycle) [39]. The mPFS (primary end-point) and OS are 11.1 (95% CI 7.6–14.7) and 17.4 (95% CI 13.4–25.7), respectively. Severe AEs reported included diarrhea, neutropenia, nausea, anemia, and thrombocytopenia.

Pralsetinib is a RET inhibitor approved after results from the ARROW study showed the safety and efficacy of this agent in metastatic non-small cell lung cancer and advanced or metastatic thyroid cancer [111]. This study sought to determine efficacy through the ORR of this agent for other cancers with RET fusions, including pancreatic, colon, CCA, and unknown primary. The ORR was 53% (CI 29–76) with 11% CR and 42% PR. This study had 3 CCA patients and 2/3 had a clinical response. The benefit of drugs targeting the DNA damage repair genes, including AT-rich interaction domain 1A (ARID1A), protein poly-bromo1 (PBRM1), and BRCA1-associated protein 1 (BAP1) in aBTCs, is under investigation (NCT03207347; NCT04042831). Ceralasertib (AZD6738), a selective ATR inhibitor that is expected to accentuate DNA damage when used with a PARP inhibitor (olaparib) and ICI (durvalumab), is being studied to treat aBTCs in the second line [112]. Adding DKN-01, a humanized mAb targeting Dickkopf-1 (DKK1), a Wnt pathway GC in the first line for aBTCs was tolerable with no dose-limiting toxicity and ORR exceeding 20% [40]. It is being studied with Nivo in a phase II study (NCT04057365). Novel agents targeting key pathways promoting carcinogenesis in BTCs, such as JAK/STAT (BBI503), Wnt/β-catenin (NCT03507998), and NOTCH (brontictuzumab), have shown promising preclinical and early-trial evidence and are expected to expand the arsenal of targeted therapies in coming years [113]

## 4. Immunotherapy in Biliary Tract Cancers

In the current clinical practice, immunotherapy can be broadly divided into ICIs and less explored adoptive cell therapy (chimeric antigen receptor T cell therapy or CAR-T) and vaccines. Reported results and ongoing trials with immunotherapy are summarized in Table 1 (above) and Table 3 (below).

### 4.1. Immune Checkpoint Inhibitors

The current ICIs can be broadly divided into three classes: (i) Cytotoxic T-Lymphocyte Antigen 4 (CTLA4) inhibitors such as ipilimumab and tremelimumab; (ii) Programmed cell death-1 (PD-1) inhibitors such as pembrolizumab and nivolumab; (iii) Programmed death-ligand 1 (PD-L1) inhibitors such as durvalumab, avelumab, and atezolizumab. PD-1 and PD-L1 inhibitors were studied alone or in combination with chemotherapy or targeted therapy, while CTLA4 inhibitors were combined with PD-1 or PD-L1 inhibitors. Pembrolizumab is recommended in patients with mismatch repair deficient (d-MMR) or microsatellite instability—high (MSI-H) and higher TMB (>10) aBTCs in the first line [114,115,116]. In one of the first reports published in 2017, 86 MSI-H/ d-MMR advanced cancer patients (12 different cancer types, including 40 colorectal cancers) were treated with pembrolizumab [114]. The ORR of the entire group was 56% (21% CR, 33% PR, and 23% SD). About 5% (4/86) of the enrolled cancers were CCA and the ORR among them was 25% (1 CR) with 100% DCR (3 SD + 1 CR). In KEYNOTE-158, MSI-H/dMMR refractory non-colorectal advanced tumors (27 types including CCAs) were treated with pembrolizumab [115]. It had 22/233 (9.3%) CCA patients and the ORR among them was 41% (2-CR and 7-PR). The PFS and OS were 4.2 months and 24.3 months, respectively. The median DOR of this cohort had not been reached at the time of publication

Nivolumab has a category 2B recommendation (per national comprehensive cancer network or NCCN guidelines) in the second line and is typically offered to patients who do not have targetable mutations and may not tolerate chemotherapy [117]. This approval was based on a trial published in 2020 where 54 refractory aBTC (>1 and ≤3 lines) patients were treated with nivolumab. Tumor samples were available for 42 patients and 18 of them (43%) expressed PD-L1. The ORR was 22% (11/46, 0-CR, 10-PR, and 1 unconfirmed PR), with 37% (17/46) having an SD based on RECIST criteria and investigator-review. The ORR and SD were 11% and 39%, respectively, on central review in this study. Interestingly, 50% (9/18) and 28% (5/18) of the patients expressing PD-L1 had an evaluable response in the investigator review and central review, respectively.

Combining ICI with chemotherapy and TKI is being studied in the first line. The TOPAZ-1 trial results (discussed above) are very encouraging and may open doors for many such combinations going forward [27]. In KEYNOTE-966 (phase III trial), the efficacy and safety of pembrolizumab and GC combination are being studied (vs. GC + placebo) [118]. Interim results of phase II trials with pembrolizumab and olaparib showed acceptable safety [119]. The combination of nivolumab and Nan-Iri/5FU did not provide the expected results in the BiT-03 trial [120]. Bintrafusp alfa, a bifunctional fusion protein targeting TGF-β and PD-L1, had promising success in the phase I trial reported in 2020 [51]. It was studied with second-line (N = 30) Asian patients with BTC (including one ampullary cancer). It was well tolerated with ≥G3 events in 37% (11/30) and G5 events in 10% (3/30). The ORR was 20% with 18 m of DOR. It is currently being studied in the first line combined with GC (GC + bintrafusp alfa vs. GC + placebo) [121].

The LEAP-005 study that evaluated the safety and efficacy of lenvatinib and pembrolizumab as second-line therapy for advanced solid tumors had 31 for BTC patients [46]. The ORR was 10% (95% CI 2–26) with DCR 68% (95% CI 49–83.) There were treatment-related AEs for 97% of patients, including 48% having grade 3–4 AEs. It was concluded that lenvatinib and pembrolizumab have some efficacy as second-line agents with tolerable side effects in patients with BTC. JS001-ZS-BC001 trial, an open-label, phase II clinical study, evaluated the efficacy and safety of toripalimab or Tor (PD-1 inhibitor) plus Gem/S-1 in the first line [36]. A total of 39 patients received this combination, with the response rate being 20.6% and DCR 85.3%. The PFS was 6.7 months, with grade 3/4 non-hematologic AEs seen in 20.5% of patients, while grade 3/4 hematologic AEs were seen in 69.2%. The study showed promising results in line with the TOPAZ trial. When Tor was combined with gemcitabine/oxaliplatin (D1 and D8, Q3W for six cycles) and lenvatinib (8 mg) in a phase II trial with locally advanced iCCA (N = 30), the ORR was 80% (1CR and 3 patients obtained enough control to allow for resection), DCR was 93.3%, PFS was 10 m, and DOR was 9.8 m. ORR was related to PD-L1 expression and DNA damage repair mutations in the tumors [122]. On the other hand, ORR and DCR were 30% and 87%, respectively, when lenvatinib plus the GemOX arm of a phase II trial reported in gemcitabine/oxaliplatin combination was presented last year [38].

In the CTEP 10139 trial, atezolizumab (Atezo) alone was compared with the combination of MEK inhibitor combimetinib (Cobi) [123]. Although the PFS was higher in the combination group (3.6 months vs. 1.9 months; *p* = 0.027), the OS and the ORR were similar in both groups. The combination of Atezo with varlilumab, a CD 27 agonist (NCT04941287), is now being studied with and without Cobi. Arginase inhibition by a novel agent (INCB001158) was well tolerated in the first line when given with GC and is being explored as another option in this area [124].

Immune-related adverse events (irAE) are organ-specific inflammatory responses invoked by ICIs similar to autoimmune diseases [16]. They can affect any organ (such as colitis, dermatitis, hepatitis, pneumonitis, hypophysitis, myocarditis, myositis, and thyroiditis) and can be life-threatening [125,126]. In the nivolumab trial, only 17% of the study population had grade 3 or grade 4 irAE [117]. In the bintrafusp alfa trial, only 37% had ≥ grade 3 irAEs [51]. The typical management for low-grade irAEs is holding the therapy and restarting it after resolution [127]. High-dose steroids and immunosuppressants such as infliximab are used in severe irAEs.

### 4.2. Chimeric Antigen Receptor T Cell Therapy and Vaccines in Biliary Tract Cancers

CAR T-cell therapy involves creating a chimeric antigen receptor (CAR) that targets an antigen in the cancer cells, allowing the host T-cells to identify the tumor cells and destroy them [128]. This approach is approved for use in various hematologic malignancies, but its use in solid tumors remains experimental [129]. The key to successful development involves determining the correct antigen to target, which is expressed in large numbers on tumor cells but is found in small numbers on healthy cells. More specific and reliable biomarkers are being studied to target CCA more effectively with CAR-T [130]. Studies with several other CAR targets, including CD133, EGFR, Integrin αvβ6, and Anti-MUC1, have shown positive results [130,131,132,133]. Of 19 patients enrolled in a study evaluating CART-EGFR, and with 17 evaluable, 1 patient saw CR for 22 months and 10 saw SD ranging from 2.5 to 15 months, with a median PFS of 4 months [134]. Another study using anti-MUC1 CAR T-cells showed significantly decreased fluorescence of MUC1 expressing cholangiocarcinoma cells after 3 and 5 days of exposure [133]. While there have been positive results in these trials, more data and larger studies are needed to further assess the safety and efficacy of CAR-T therapy in CCA.

Additionally, vaccines have been developed from various peptides such as MUC1, WT1, and other combinations to mount an immune response against the patient’s cancer. A three-peptide vaccine consisting of cell division cycle associated 1 (CDCA1), cadherin 3 (CDH3), and kinesin family member 20A (KIF20A) showed response in 5 of 9 patients enrolled, with a median PFS of 3.4 months and OS of 9.7 months [135]. A study using MUC1 peptide showed a response in only 1 of 8 patients enrolled but had a tolerable side effect profile [136]. Another study used lymphocyte antigen 6 complex locus K, TTK protein kinase, insulin-like growth factor-II mRNA-binding protein 3, and DEP domain containing 1, with 7 of 9 patients showing response and producing a median PFS of 5.2 months and OS of 12.7 months [137].

## 5. Systemic Therapy in Early-Stage Biliary Tract Cancers

Capecitabine is the preferred agent for AT in BTCs based on the BILCAP trial [138]. On the other hand, BCAT and PRODIGE 12 trials could not show the clinical benefit of gemcitabine or gemcitabine/oxaliplatin combination over observation [139,140,141]. A recently presented pooled analysis of these two trials further proved this point [142]. A total of 419 patients were included in the two studies, which showed no difference in PFS (2.9 years in gem-based vs. 2.1 years in observation; HR = 0.91; *p* = 0.45) or OS (5.1 years vs. 5 years; HR = 1.03; *p* = 0.83). Radiation alone (XRT) or chemoradiation (CRT) in the adjuvant setting is not a popular approach in managing BTC. CRT is offered to eCCA and GBC patients with positive margins or lymph nodes [143,144,145]. Retrospective studies showed benefits with chemotherapy only in resected BTCs, but it is difficult to compare the AT strategies as CRT or XRT is offered to BTCs with high-risk factors (positive margins/lymph nodes) [146].

Neoadjuvant (NAT) systemic therapy is not a standard approach in resectable BTCs. Some case reports and retrospective studies show the benefit of NAT downstaging the locally advanced or unresectable BTCs enough to have resection [147,148,149]. The addition of pre-operative radiation can increase the probability of R0 resection in these tumors [150,151]. On the other hand, NAT did not result in any survival advantage in managing resectable BTCs in the reported studies [152]. Multiple trials investigating the role of neoadjuvant therapy in resectable (GC-D in NCT04308174 or DEBATE; GC in NCT03673072; GC/NP in NCT03579771) and unresectable/locally advanced BTCs (FOLOXIRI in NCT03603834; toripalimab + GEMOX + lenvatinib in NCT0450628) are underway that may give us a definite answer in the coming years. In the current practice, systemic options typically for NAT are similar to those used for treating aBTCs (such as GC).

Locoregional therapy (LRT) with high-dose XRT (58–67.5 Gy in 15 fractions) and SBRT (30–50 Gy in 3 to 5 fractions) improves local control and OS in unresectable iCCA, and can be an option for suitable patients [153,154]. Other LRTs such as transcatheter arterial chemoembolization (TACE) and transarterial radioembolization (TARE) are not typically employed in treating BTCs. SBRT plus capecitabine combination increased local control rates (≈80%) with minimal toxicity (no ≥ grade 3 toxicity) in unresectable perihilar CCA [155]. Other trials intended to see the benefit of SBRT and chemotherapy combinations were closed due to low accrual (NCT01151761 and NCT00983541). ICI with TACE or SBRT, or TARE trials, are underway (NCT03898895, NCT04866836, NCT03937830, NCT02821754, NCT04238637, and NCT04708067), which may open up more options in the near future.

## 6. Conclusions

GC has been the standard of care for first-line treatment of BTCs for more than a decade. It took considerable time to compile the present arsenal of therapeutic options to treat this lethal cancer (illustrated in the graphical abstract). Currently, the benefit of adding NP to it or replacing gemcitabine with NUC-1031 is being studied. Second-line systemic therapy for patients ineligible for targeted therapy is limited. Although the NIFTY trial suggested the benefit of Nan-Iri/5FU, FOLFOX is typically used. The success of FGFR2, IDH, NTRK, and BRAF (V600E) inhibitors in the second line is remarkable, but their use is restricted by the low prevalence of the respective targets in BTCs. Therapy directed at new targets such as VEGF, HER2, and RET are being studied and may open doors for new options. Pembrolizumab is preferred in MSI-H patients, with little evidence for the use of nivolumab in second-line MSS patients. The TOPAZ-1 trial results may be a game-changer and can bring ICI into the first line.

The landscape of BTC management has started to change in recent years. The oncology practice is moving away from traditional chemotherapy to personalized medicine. The accessibility to tumor mutation profiling and circulating tumor DNA or ctDNA genomic profiling contributed to the success of targeted therapy and paved the way for many new agents. By studying the ongoing trials, it is clear that the focus is also on expanding the use of ICIs with chemotherapy (GC or GEMOX) or selected targeted therapy (rucaparib, DKN-01, entinostat). Such combinations will cater to broader BTC populations as the prevalence of mutations does not restrict eligibility. Other investigational therapies such as CAR-T therapy and vaccines are driving the advancement in treatment options and patient outcomes. With the continued success of clinical trials, these agents could be seen in the near future to join the fight against BTC and provide a much-needed breath of fresh air to the treatment options that we can offer our patients.

## Figures and Tables

**Figure 1 cancers-14-02137-f001:**
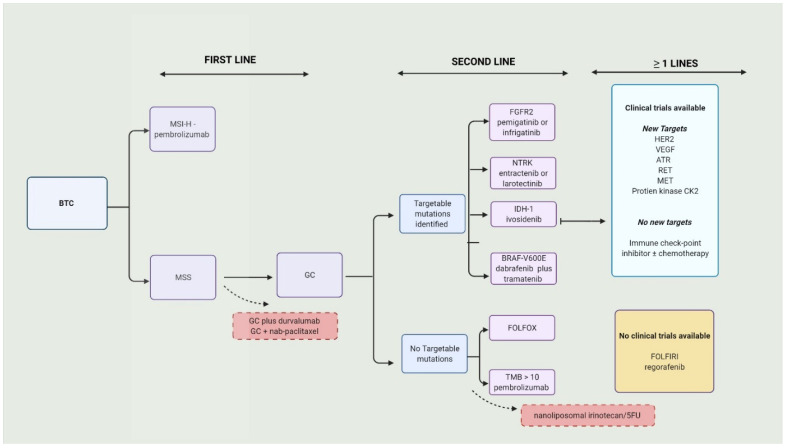
Current approach to biliary tract cancers. BTC—biliary tract cancers; MSI-H—microsatellite instability; MSS—microsatellite stable; GC—gemcitabine/cisplatin; FGFR2—fibroblast growth factor 2; IDH—isocitrate dehydrogenase-1; NTRK—neurotrophic tyrosine receptor kinase; HER2—human epidermal growth factor receptor 2 inhibitors; VEGF—vascular endothelial growth factor; TMB—tumor mutational burden; ATR—ataxia telangiectasia mutated and Rad3-related.

**Table 2 cancers-14-02137-t002:** Ongoing trials with targeted therapy in biliary tract cancer.

Line	Phase	Clinical Trial Identifier	Target of the Drug	Treated Cancer Group	Experimental Arm	Comparative Arm	Primary Outcome	Secondary Outcome (Main)
First line	III	NCT03773302	FGFR rearrangement	CCA	Pemigatinib	GC	PFS	OS, OR, DOR, DCR
	III	NCT03773302	FGFR2 fusion/translocation	CCA	Infrigatinib	GC	PFS	OS. DCR, DOR, BOR
	III	NCT04093362	iCCA with FGFR2	iCCA	Futibatinib	GC	PFS	ORR. DCR. OS. Safety/Tolerability
	II	NCT03768414	Not specific	BTC	GC/NP	GC	OS	PFS, ORR, DCR
	II	NCT03579771	High risk *	Resectable IHC	GC/NP	None	SR	RR, R0; OS; PFS
Subsequent lines k	II	NCT04722133	HER 2	aBTC	Trastuzumab-pkrb + FOLFOX	None	ORR	PFS, OS, DCR, incidence of TRAE
	II	jRCT2031180150	HER 2	Advanced solid tumors ^#^	Trastuzumab and pertuzumab	None	ORR	PFS, OS, DoR, safety
	II	NCT02091141 (My Pathway)	HER 2	BTC ^#^	Trastuzumab and pertuzumab	None	ORR	DCR, PFS, OS, AE
	II	NCT04466891	HER 2	BTC	Zanidatamab monotherapy	None	ORR	DoR; DoR > 16 wks; DCR, PFS, OS; incidence of TRAE, PK
	II	NCT02999672	HER 2	CCA ^#^	Trastuzumab emtansine	None	BOR	PFS, OS, TRAE, SAE, PK
	II	NCT04482309	HER2	BTC ^#^	Trastuzumab deruxtecan	None	ORR	DOR, DCR, PFF, OS, AEs, PK and immunogenicity
	II	NCT03839342.	Non-V600E BRAF mutations	Advanced solid tumors ^#^	Bimimetinib + encorafenib	None	ORR	Safety, DCR, PFS
	II	NCT02428855	IDH1 mutation	iCCA	Dasatinib	None	ORR	PFS, OS, TRAE
	II	NCT02675829	HER2 amplification	Advanced solid tumors ^#^	Ado-Trastuzumab emtansine	None	ORR	None
	II	NCT03207347	BAP1 and other DDR genes	CCA ^#^	Niraparib	None	ORR	PFS, OS, TRAE
	II	NCT03212274	IDH1/2 mutation	CCA	Olaprib	None	ORR	PFS, OS, safety
	II	NCT04042831	DNA repair gene mutation	BTC	Olaparib	None	ORR	OS, PFS, TRAE, DoR
	II	NCT03207347	DNA repair gene mutation	CCA ^#^	Niraparib	None	ORR	OS, PFS, TRAEs
	II	NCT02162914	VEGF mutation	CCA	Regorafenib	None	PFS	RR, OS
	II	NCT03339843	CDK 4/6 mutation	CCA ^#^	Abemaciclib	None	Anti-tumor activity	PFS, OS, toxicity
	II	NCT04003896	CDK 4/6 mutation	BTC	Abemaciclib	None	ORR	PFS, DCR, OS, QoL
	II	NCT02232633	STAT3 inhibitor	CCA	BBI503	None	DCR	ORR, OS, PFS, PK TRAE
	II	NCT03878095	IDH1/2 mutation	CCA ^#^	Ceralasertib + olaparib	None	ORR	PFS, OS, DoR, Safety
	I/II	NCT02273739	IDH2 mutation	Advanced solid tumors ^#^	Enasidenib Enasidenib	None	DLT, ECOG	Plasma concentration metrics
	I	NCT04764084	HRR mutations	CCA ^#^	Niraparib + anlotinib	None	DLT, MTD	ORR, PFS
	I	NCT04521686	IDH1 R132-mutant advanced solid tumor types or circulating tumor DNA IDH2 R140 or IDH2 R172 mutation (CCA)	CCA ^#^	LY3410738 LY3410738 + GC		Maximum tolerated dose	ORR Safety and tolerability Efficacy PK properties
	I	NCT02381886	IDH1 mutation	BTC ^#^	IDH305	None	DLT	TRAE, PK, delta 2-hydroxyglutarate, ORR, SAE
	I	NCT03272464	BRAF-V600E	BTC ^#^	JSI-1187 + dabrafenib	None	TRAE	DOR, OS, PFS, TTP
	I	NCT04190628	BRAF-V600E	BTC ^#^	ABM-1310 + cobimetinib	None	MTD	TRAE, PK, DOR, OS, PFS, TTP
	I	NCT02451553	No specific target	BTC ^#^	Afatinib dimaleate + capecitabine	None	AE, DLT, MTD	DOR, OS, PFS, RR, TTP, biomarker profile
	I	NCT03507998	Wnt/β-catenin signaling inhibitors	BTC ^#^	CGX1321	None	TRAE	PK

^#^ Basket trial; * T-stage ≥ Ib (Ib-IV); solitary lesion > 5 cm; Multifocal tumors or satellite lesions present; BTC—biliary tract cancers include gall bladder cancers and CCA; iCCA—intrahepatic cholangiocarcinoma; eCCA—extra-hepatic cholangiocarcinoma; CCA—cholangiocarcinoma includes iCCA and eCCA; FGFR2—fibroblast growth factor 2; IDH—isocitrate dehydrogenase-1; VEGF—vascular endothelial growth factor; HER2—human epidermal growth factor receptor 2 inhibitors; STAT—signal transducer and activator of transcription; GC—gemcitabine/cisplatin; DCR—disease control rate; ORR—objective response rate; BOR—best overall response; DOR—duration of response; TTP—time to progression; SR—surgical resect ability; TRAEs—treatment-related adverse events; SAE—serious adverse events; PK—pharmacokinetics; RR—response rate; DLT—dose limiting toxicity MTD—maximum tolerated dose; QoL—quality of life; BOR—best overall response.

**Table 3 cancers-14-02137-t003:** Ongoing trials with immunotherapy in biliary tract cancer.

Line	Phase	Clinical Trial Identifier	Treated Cancer Group	Experimental Arm	Comparative Arm	Primary Outcome	Secondary Outcome (Main)
First line	III	NCT04003636	BTC	Pembrolizumab + GC	GC + placebo	OS	PFS, ORR, DOR
	II/III	NCT04066491	BTC	Bintrafusp alfa	GC + placebo	OS DLT	PFS, DOR, ORR
	II	NCT04217954	BTC	HAIC (oxaliplatin + 5-FU) + toripalimab (T) + bevacizumab	None	PFS, ORR	OS, AE, CA 19-9, DCE-MRI signal change, DWI MRI signal change
	II	NCT04172402	BTC	TS-1 + gemcitabine + nivolumab	None	ORR	None specified
	II	NCT03898895	iCCA	Camrelizumab + radiotherapy	GC	PFS	OS, AE, tumor response
	III	NCT03478488	BTC	KN035 (PD-L1 antibody) + gemcitabine + oxaliplatin	GEMOX	OS	PFS, ORR, DCR, DOR, TTP
	II	NCT03796429	BTC	Gemcitabine/S-1 + toripalimab	None	PFS, OS	ORR, Safety
	II	NCT04027764	BTC	Toripalimab + S1 and albumin paclitaxel	None	ORR	PFS, DCR, OS
	II	NCT04191343	BTC	Toripalimab + GEMOX	None	ORR	None specified
	II	NCT04300959	BTC	Anlotinib hydrochloride + PD1 + gemcitabine + cisplatin	Gemcitabine Cisplatin	OS 1 yr	OS 2 yr, PFS, ORR, AE
Subsequent lines	II	NCT03482102	HCC, BTC	Tremelimumab + durvalumab + radiation	None	ORR	AE, OS, DCR, PFS, DOR, TTP
	II	NCT04238637	BTC	Durvalumab (D) vs. D + T	None	ORR	Safety, DoR, PFS, OS
	II	NCT02821754	HCC, BTC	D + T	D +T + TACE D + T + RFA D + T + Cryo	PFS	Safety
	II	NCT02703714	BTC	Pembrolizumab and sargramostim (GM-CSF)	None	ORR	AE, PD-L1 positivity, PFS, OS, DOR
	I/II	NCT03937895	BTC *	Allogeneic natural killer cells + pembrolizumab	None	Phase I—DLT Phase II—ORR	TTP, toxicity
	II	NCT04306367	BTC	Pembrolizumab and olaparib	mFOLFOX-historical control	ORR	DOR, PFS, OS, safety
	II	NCT04295317	iCCA—adjuvant	PD-1 blocking antibody SHR-1210 + capecitabine	None	PFS	OS, side effects
	II	NCT03250273	BTC, PDA	Entinostat + nivolumab	None	ORR	Toxicity, PFS, OS, DOR
	II	NCT02866383	BTC, PDA	Nivolumab + ipilimumab + radiotherapy	Nivolumab + radiotherapy	CBR	AE, ORR, PFS, OS, QOL
	II	NCT04057365	BTC	DKN-01 + nivolumab	None	ORR	PFS, OS
	II	NCT03639935	BTC	Rucaparib + nivolumab	None	4-month PFS rate	Response rate, PFS, OS
	II	NCT04299581	iCCA	Camrelizumab + cryo	None	ORR	DOR, PFS, OS, DCR, AE
	II	NCT03999658	BTC ^#^	STI-3031 anti-PD-L1 antibody	None	ORR	DOR, CR, PFS, 1-year PFS rate, correlative studies
	II	NCT03801083	BTC	Tumor infiltrating lymphocytes (TIL) + aldesleukin	None	ORR	CRR, DOR, DCR, PFS, OS, QOL
	I/II	NCT03684811	BTC ^#^	FT-2102 vs. FT-2102 + nivolumab	None	DLT, Dose, ORR	ORR, AE, PFS, TTP, DOR, OS, TT
	I/II	NCT03475953	BTC ^#^	Regorafenib + avelumab	None	I = dose II = antitumor activity	MTD, DLT, toxicity, AE, PK and correlative studies
	I/II	NCT03785873	BTC	Nal-Irinotecan + nivolumab + 5-Fluorouracil + leucovorin	None	I = DLT II = PFS	AE, ORR, OS
	I	NCT03849469	iCCA ^#^	XmAb^®^22841 and pembrolizumab	XmAb^®^22841 Monotherapy	Safety and tolerability	None
	I	NCT03257761	BTC, PDA, HCC	Guadecitabine + durvalumab	None	AE, Tumor response	OS, PFS

BTC—biliary tract cancers include gall bladder cancers and CCA; iCCA—intrahepatic cholangiocarcinoma; eCCA—extra-hepatic cholangiocarcinoma; CCA—cholangiocarcinoma includes iCCA and eCCA; PDA—pancreatic cancer; HCC—hepatocellular cancer; FGFR2—fibroblast growth factor 2; IDH—isocitrate dehydrogenase-1; VEGF—vascular endothelial growth factor; HER2—human epidermal growth factor receptor 2 inhibitors; HHR—homologous recombination repair; GC—gemcitabine/cisplatin; GM-CSF—granulocyte-macrophage colony-stimulating factor; TACE—transcatheter arterial chemoembolization; RFA—radiofrequency ablation; Cryo—cryotherapy; HAIC—hepatic arterial infusion chemotherapy; CPS—combined positive score; MSI-H—microsatellite instability; DCE—dynamic contrast enhanced; DWI—diffusion weighted imaging; TTP—time to progression; CBR—clinical benefit rate; QOL—quality of life; TTR—time to response; ^#^—basket trials with BTC among them; * at least 1% CPS PD-L1 or MSI-high or dMMR positive.

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
