# Peer review of "Changing Landscape of Systemic Therapy in Biliary Tract Cancer"

_cancers, 2022, doi:10.3390/cancers14092137_

Round 1
Reviewer 1 Report
Woods et. al., described about the shortcomings of the current treatment methods available in the BTC and the need for more effective combination or targeted therapies in order to improve the survival rate of the patients along with less adverse effects. The authors have tried to touch base all the current therapies and the clinical studies. While the similar line of studies reported already, the current article has interesting observation which is beneficial to researchers in cancer studies. I had few suggestions as following:
- Since phase III (ABC-02) is the extension of phase II (ABC-01) clinical trails with improved tumor control and progression-free survival, the authors are advised to mention details about the study and the adverse effects of the combination.
- Authors are suggested to include the references in all, the tables.
- Authors are advised to mention the combination therapy of gemcitabine, cisplatin and S-1 (an oral fluoropyrimidine derivative) as a standard regimen for advanced cases.
- Authors are advised to mention the adverse effect of all the therapies or combination drugs wherever applicable.
- Authors are suggested to mention ClarIDHy study, since it was another positive phase III study that showed the efficacy of second-line chemotherapy
- Authors are advised to include mutations of chromatin-remodeling genes, such as AT-rich interaction domain 1A (ARID1A), protein poly-bromo1 (PBRM1), and BRCA1-associated protein 1 (BAP1) in targeted therapy
- Authors are suggested to include important combination drugs and studies in HER2 alterations such as combination of trastuzumab and pertuzumab in the MyPathway trial (NCT02091141), ongoing trials are evaluating afatinib (NCT02451553) and trastuzumab–emtansine (NCT02999672)
- Authors are suggested to include the studies relates to BRAF mutations such as Vemurafenib, in combination with an MEK inhibitor, Selumetinib, an inhibitor of MEK1/2 proteins, combination of solumetinib with gemcitabine
- Authors are suggested to include the promising targets for potential inhibitors such as proteins of the JAK/STAT, Wnt/β-catenin, Hedgehog and Notch signalling pathways, inhibitors for mutations in chromatin-remodelling genes, such as ARID1A, PBRM1, and BAP1, and inhibitors of cancer-associated fibroblasts
- Authors are suggested to discuss the locoregional therapy as a promising strategy
Author Response
Please find the response to all the comments/suggestions attached.
Thanks
Ashish Manne

Reviewer 2 Report
The authors present a comprehensive review of recently published and ongoing trials in the field of biliary tract cancer. Given the many recent updates in this disease a review such as this is an important contribution to the field and the authors spent a considerable amount of time putting this manuscript together. The review is generally well organized and covers most of the major recent updates in biliary tract cancers however there are several areas of the manuscript that are lacking or in need of clarification as noted in the comments below.
Abstract
Summarizes current state of management and future directions well
Introduction
The reported distribution of BTC lesions you cite does not include gallbladder cancer despite your intro stating you are including gallbladder cancer in your definition of BTC.
Recent data also shows that the incidence of intrahepatic CCA may be rising due to a number of reasons (intrahepatic CCA likely represents much higher than 10% of CCAs). This may be worth commenting on as we know that there are differences in the prevalence of targetable mutations in intrahepatic vs extrahepatic CCA.
Set up is good but need a statement explaining that for the vast majority of patients palliative systemic therapy is the only option. Need a better segue into the last sentence of the introduction.
Section 2.1
Line 70 - would classify aBTC as unresectable “or” metastatic not “and”
Paragraph 1 - Would give the date of publication of ABC02 for context and state the outcome data briefly that led to it establishing gem/cis as current first line standard of care. Going into more detail about this trial in a standalone first paragraph may be helpful as the majority of the trials you discuss subsequently are being compared to this trial.
Line 79 to 80 – This statement is misleading and out of context. It should be moved to your discussion regarding second line therapies. The phase II trial you cite here does not provide any evidence that FOLFIRINOX should be considered a first line option especially given the negative results of PRODIGE 38.
Line 94 – This sentence is confusing as written. Would say “about half of the patients enrolled in this trial” not “them” and specify why this dose change was made if possible. Would also be helpful to clearly state this was a first line trial.
Line 99 – Would include actual data here if you are going to make these comparisons or reference your supplementary table which includes this data here. Would also remove neuropathy from this statement or modify the wording as neuropathy rates were not reported in the ABC02 trial but one would infer from this sentence that you are comparing the rates of neuropathy in both trials.
Would also consider stating either here or in the next paragraph that the phase III SWOG1815 has completed accrual and once published will help answer the question of GAP vs gem/cis.
Line 111 - Need to make it clear that this data has only been presented in abstract form thus far. Although promising and I agree that this regimen is likely to become standard of care, we are still awaiting final publication of this trial and should not make any strong conclusions until the full data is available for review. Additionally, the data presented at ASCO is not without limitations/caveats. The survival curves do not separate until after 6 months raising the question of whether the results are driven by maintenance durvalumab and not the combination of chemo/immunotherapy during the first 6 months. While there are clearly some patients that are long term responders as evidenced by the tail of the survival curve the median OS difference of only 1.3 months shows we still need new therapies to meaningfully change outcomes in this disease and need to be better able to identify the subset of patients that will respond to immunotherapy.
Line 121 – Supplementary Table S1 is a useful comparison however it may be easier to read if the title or author were added to the first two columns similar to how TOPAZ1 is listed in the last column
Section 2.2
Line 133 – Would consider commenting on why this is relevant. Theory is that platinum sensitive disease in the first line selects for response to 2L folfox?
Line 135 – With this sentence it is unclear if the similar outcomes you are referring to pertain to the outcomes of FOLFOX in the ABC06 trial (which I would also name above) or that outcomes between gem, cape, or gem cape are similar.
Line 146 to 150 – incomplete sentence
Section 3.1
FGF is referred to as “fibroblast” not “fibroblastic” growth factor
Line 194 – Do you have a reference to support mutual exclusivity between RAS/RAF mutations and FGFR fusions?
Line 195 to 202 – Lumping both the infigratinib and pemigatinib studies together is reasonable given their similarity however due to practice changing nature of these studies it may be helpful to describe the trials individually in more detail as you did with the chemotherapy trails above.
Line 206 – not clear which of the trials you are referencing here.
Section 3.2
Line 214 to 216 – Would be helpful to further expand on how the lack of TMB/MSI/PDL1 positivity indicates a “profound effect of IDH pathway” as this is not entirely clear.
Section 3.3
Would be helpful to add a sentence or two discussing the prevalence of NTRK fusions in BTCs
Section 3.4
May consider an intro sentence or two describing the role of VEGF in BTC tumorigenesis similar to the previous 3 sections as justification for the trials you subsequently describe.
Section 3.5
The citation numbers are not associated with the correct studies within this paragraph. Would review citations throughout manuscript to ensure all are correct.
Line 266 – I assume here by ‘it” you mean HER2 targeted therapy but would clarify the wording. Additionally, HER2 therapy has been standard of care in gastroesophageal cancers for many years thus I would not describe it as gaining attention in gastric cancer. Agree however that the role of HER2 directed therapy in colon cancers is still developing. This sentence may also fit better at the end of the prior paragraph.
Line 277 – Would clarify what you mean by a popular target for treatment. The statement re percentage of HER2 amplified BTCs would probably also fit better within the preceding paragraph describing role of HER2 in tumorigenesis.
Line 287 – Ongoing trial involving the antibody drug conjugate trastuzumab deruxtecan should be mentioned here as well. This drug has promising activity in breast and gastric cancer.
Section 3.6
Line 302 – Agree that these are promising PFS/OS but would caution against stating the PFS and OS are impressive compared to GC given that the silmitasertib trial you are describing was a non-randomized phase I/II trial that did not directly compare to GC alone.
Line 309 to 310 – This sentence is incomplete
Section 4.1
Line 344 to 353 – You provide the stats from several important IO studies here but do not introduce them beyond providing a reference. It would be helpful to readers to at least introduce the name of each trial and describe what the population/intervention was before providing the ORR, DCR, etc
Line 392 – This paragraph seems beyond the scope of your intentions with this review and if needed due to space constraints could be omitted to adequately address the comments above or that of other reviewers.
Section 5
Line 441 and 442 – Devoting only a single sentence to neoadjuvant treatment strategies without any references seems inadequate here. There are numerous retrospective studies and several ongoing clinical trials investigating neoadjuvant therapy in biliary tract cancers. While possibly outside of the scope of your review neoadjuvant strategies should at least be mentioned as an area of high interest especially given the lack of benefit seen in adjuvant therapy trials that you mention previously.
Tables – Would make sure all tables are formatted using a similar style. For example, phase is in a different location in table 1 and 2. The column for NCI# is listed as “trial number” in table 1 and as “clinical trial” in table 2.
Author Response

(The authors gave the same response as above.)

Round 2
Reviewer 2 Report
Authors have addressed my comments adequately.